# Echocardiographic Screening for Rheumatic Heart Disease in a Ugandan Orphanage: Feasibility and Outcomes

**DOI:** 10.3390/children9101451

**Published:** 2022-09-23

**Authors:** Massimo Mapelli, Paola Zagni, Valeria Calbi, Laura Fusini, Aliku Twalib, Roberto Ferrara, Irene Mattavelli, Laura Alberghina, Elisabetta Salvioni, Cyprian Opira, Jackson Kansiime, Gloria Tamborini, Mauro Pepi, Piergiuseppe Agostoni

**Affiliations:** 1Centro Cardiologico Monzino, IRCCs, Via Parea 4, 20138 Milan, Italy; 2Department of Clinical Sciences and Community Health, Cardiovascular Section, University of Milan, 20122 Milan, Italy; 3Terapia Intensiva Neonatale, Ospedale Fatebenefratelli P.O. Macedonio Melloni, Via Macedonio Melloni 52, 20129 Milan, Italy; 4San Raffaele Telethon Institute for Gene Therapy (SR-TIGET), IRCCS San Raffaele Scientific Institute, Via Olgettina, 60, 20132 Milan, Italy; 5Pediatric Immunohematology Unit and BMT Program, IRCCS San Raffaele Scientific Institute, Via Olgettina, 60, 20132 Milan, Italy; 6Division of Paediatric Cardiology Uganda Heart Institute, Mulago Hospital and Complex, Kampala P.O. Box 37392, Uganda; 7Molecular Immunology Unit, Medical Oncology Department—Department of Research, Fondazione IRCCS Istituto Nazionale dei Tumori, 20132 Milan, Italy; 8Department of Neurorehabilitation Sciences, Istituto Auxologico Italiano, IRCCS, 20132 Milan, Italy; 9Hospital Lacor, Gulu P.O. Box 180, Uganda

**Keywords:** rheumatic heart disease, rheumatic fever, mitral valve, echocardiographic screening, developing countries

## Abstract

Background: Rheumatic heart disease (RHD) is a major cause of cardiovascular disease in developing nations, leading to more than 230,000 deaths annually. Most patients seek medical care only when long-term structural and hemodynamic complications have already occurred. Echocardiographic screenings ensure the early detection of asymptomatic subjects who could benefit from prophylaxis, monitoring and intervention, when appropriate. The aim of this study is to assess the feasibility of a screening program and the prevalence of RHD in a Ugandan orphanage. Methods: We performed an RHD-focused echocardiogram on all the children (5–14 years old) living in a north Ugandan orphanage. Exams were performed with a portable machine (GE Vivid-I). All the time intervals were recorded (minutes). Results: A total of 163 asymptomatic children were screened over 8 days (medium age 9.1; 46% male; 17% affected by severe motor impairment). The feasibility rate was 99.4%. An average of 20.4 exams were performed per day, with an average of 15.5 images collected per subject. Pathological mitral regurgitation (MR) was found in 5.5% of subjects, while at least two morphological features of RHD were found in 4.3%, leading to 1 “definite RHD” (0.6%) case and 13 “borderline RHD” cases (8.1%). Six congenital heart defects were also noted (3.7%): four atrial septal defects, one coronary artery fistula and one Patent Ductus Arteriosus. Conclusions: We demonstrated the feasibility of an echocardiographic screening for RHD in an orphanage in Uganda. A few factors, such as good clinical and hygienic care, the availability of antibiotics and closeness to a big hospital, may account for the low prevalence of the disease in our population.

## 1. Introduction

Rheumatic heart disease (RHD) is a major cause of cardiovascular disease in developing nations, leading to more than 230,000 deaths every year [1,2]. Although obtaining reliable data, especially in rural areas, can be particularly complex, in Uganda RHD represents the most common cause of heart disease in young adults [3,4]. RHD is a chronic process, characterized by progressive damage of the heart valves (more often affecting the mitral valve (MV)) following repeated acute rheumatic fever episodes. The autoimmune reaction triggered by the Group A streptococcus bacterial infection, usually affecting the throat, represents the *primum movens* of the disease. Over a period of months or years, these recurrent episodes produce progressive lesions at the level of the valve structures that become irreversible over the time. Heart failure, stroke, and atrial fibrillation are the most common cardiac complications of RHD. Being a disease predominantly affecting young children or adolescents, this implies a high rate of premature morbidity and mortality. In parallel with socioeconomic development and improved sanitation, RHD has gradually become rarer in high-income, while still affecting millions of people in low-income countries. Therefore, in Uganda, as it is also reported in many other underserved countries, most patients seek medical care only at the later stages of the disease, when structural and hemodynamic complications have already occurred [5]. This is also due to the long latency phase of the RHD disease in which the valve lesions, which can be well visualized anatomically but not yet able to give functional impact, are completely asymptomatic (no dyspnea, chest pain, arrhythmias). Despite the huge number of affected patients, there is still a lack of evidence about how to control RHD in different settings. The key point is that, in the absence of a proper prophylaxis, RHD patients are at a high risk of recurrent attacks of rheumatic fever, resulting in ongoing inflammation, leading to progressive fibrosis and consequential valvular damage [6,7]. Therefore, echocardiographic screenings ensure the early detection of asymptomatic individuals who could benefit from prophylaxis, monitoring, and intervention, when appropriate. Although several RHD screening programs have been run in recent decades, most of the eligible children, particularly those confined to the poorest areas, have no real opportunity to be screened.

The aim of the study is to assess the feasibility and the prevalence of RHD in a screening program in a Ugandan orphanage located in one of the poorest regions of the country.

## 2. Materials and Methods

We performed an RHD-focused echocardiogram on all the children (5–14 years old) living in an orphanage in northern Uganda (St. Jude Children’s Home, Gulu Municipality, Gulu District). This study was conducted in March 2016. St. Jude Children’s Home is a Catholic non-profit organization founded in 1987 with the specific mission to care for orphaned, abandoned or vulnerable children with different disabilities. Exams were performed with a portable machine (GE Vivid-I) (GE, Boston, MA, USA). All the time intervals (seconds and/or minutes) and number of the clips and/or still images collected were recorded (minutes).

The screening was conducted according to the World Health Organization (WHO) criteria for echocardiographic diagnosis of RHD [8]. In brief, a diagnosis of “definite RHD” required at least one (either A, B, C, or D) of the following features: A) pathological mitral regurgitation (MR) and at least two morphological features of RHD of the MV; B) mitral stenosis (MS), mean gradient ≥4 mmHg; C) pathological aortic regurgitation (AR) and at least two morphological features of RHD of the aortic valve (AV); D) borderline disease of both the AV and MV. On the contrary, the following criteria (either A, B, or C) were applied to define a “borderline RHD”: A) at least two morphological features of RHD of the MV without pathological MR or MS; B) pathological MR; C) pathological AR.

The WHO criteria [8] also define the morphological features of RHD on the MV (4 features: anterior MV leaflet thickening ≥3 mm; chordal thickening; restricted leaflet motion; excessive leaflet tip motion during systole), and on the AV (4 features: irregular or focal thickening; coaptation defect; restricted leaflet motion; prolapse).

Finally, the WHO states the criteria needed to define a pathological MR (all 4 Doppler criteria must be met: (A) seen in 2 views; (B) in at least 1 view, jet length ≥ 2 cm; (C) peak velocity ≥ 3 m/s for 1 complete envelope; (D) pansystolic jet in at least 1 envelope) and a pathological AR (all 4 Doppler criteria must be met: (A) seen in 2 views; (B) in at least 1 view, jet length ≥ 1cm; (C) peak velocity ≥ 3 m/s in early diastole; (D) pandiastolic jet in at least 1 envelope).

All cases were evaluated by 2 experienced echocardiography operators and, in case of discrepancy, by a third operator was consulted.

## 3. Results

A total of 163 asymptomatic children were screened over eight consecutive days (medium age: 9.1 ± 3.3 years; 46% male). Twenty-eight subjects (17%) were affected by severe motor impairment. None of the subjects had a previous diagnosis of heart disease (including RHD). Only in one subject was it not possible to obtain echocardiographic images appropriate for the measurements due to extreme alteration in the thoracic conformation (feasibility rate: 99.4%).

An average of 20.4 ± 7.4 exams were performed per day with an average of 15.5 ± 5.5 images collected per subject (Figure 1).

Figure 2 shows the results of the RHD screening. None of the children presented with a pathological AR, while a pathological MR was found in 5.5% of the subjects. Considering RHD anatomical features, at least two morphological features of RHD were found in 4.3% of the population. These data combined led to a total number of 1 “definite RHD” (0.6%) and 13 “borderline RHD” cases (8.1%) (Figure 3).

Figure 4 shows an example of borderline RHD and one of definite RHD.

As mentioned in Figure 3, in all the 14 children with abnormal findings (1 “definite RHD” and 13 “borderline RHD”) a follow-up evaluation with echocardiography was performed at 6 months, confirming the previous findings with no significant disease progression (none of the “borderline RHD” subjects met the criteria for “definite RHD”). Therefore, a longer follow-up was planned.

Of note, six previously unknown congenital heart defects were also noted (3.7%): four atrial septal defects, one coronary artery fistula and one Patent Ductus Arteriosus.

## 4. Discussion

In the present study, we described the feasibility of an RHD screening program in the setting of a low-income country. One of our most significant data was obtained over the short duration of the screening procedures. In fact, we were able to scan a total of 163 children in about 19 h, with an average duration of about 7 min per patient and collecting a mean of 15.5 ± 5.5 images. New-generation echocardiographic equipment allows a high volume of examinations to be performed in a short time, thanks to increasingly faster and better performing software. In addition, a selected and young population such as this one does not offer those traditional obstacles that characterize the performance of complex echocardiographic examinations, for example, in an intrahospital setting. In the hospital, in fact, pathological subjects are more frequently suffering, such as from clinically relevant valvulopathies or cardiomyopathies. This, together with the time required for bureaucratic and logistical procedures (i.e., admission office, reaching the outpatient clinic, the need to create a report), in addition to the anxiety of the patient (a child, as in this case) related to being outside his environment, is likely to significantly increase the duration of examinations. An additional factor in favor of running screening programs “on site” (in villages or, as in this case, in a community) is the increased adherence of the subjects (in our study, 100% of the children living in the orphanage and about 50 children who came by word of mouth). This finding, relevant in all communities, is even more common in a developing country where travel is often long and complex and accompanying a child to the hospital means lost time and income for a family member [9]. On the other hand, although located in a region of Uganda (Gulu district) that is still very rural, we conducted this study focusing on subjects who lived in a single facility in the surroundings of a large city (Gulu). Therefore, logistics (e.g., transporting machines, gathering people, capillarity of the intervention, adherence to the screening campaign) were facilitated. While limiting the generalization of these results to other situations (e.g., small villages far from population centers and with low socioeconomic and/or cultural status), in these regions word of mouth is very effective, being a valid instrument to recruit volunteers and, albeit with difficulty, logistical obstacles can likely be overcome. However, it is likely that a study conducted in more rural areas may involve longer lead times.

RHD often presents with moderate-to-severe multivalvular disease (63.9%), heart failure (33.4%), pulmonary hypertension (28.8%), atrial fibrillation (21.8%), stroke (7.1%), or infective endocarditis (4%) [10]. In the context of developing countries, such late presentations allow only few options for intervention, as cost and lack of access extremely limit surgical and catheter-based treatments [11,12]. Thus, due to the difficulties in RHD primary (and primordial) prevention, a screening approach based on early case detection and a successful therapeutic strategy seems to be the only feasible one.

The high sensitivity of echocardiography led the clinicians and the researchers to discover early morphological valvular changes before any clinically detectable functional lesion had developed. RHD can be considered a “cumulative” progressive disease. Therefore, younger patients diagnosed in the early stages of their disease benefit the most from this earlier diagnosis and institution of secondary prophylaxis with penicillin. However, in our study, we were able to identify only one subject (0.6%) who met the WHO criteria for “definite RHD”. This prevalence is lower than that reported in several previous larger studies, including some conducted in Uganda [13,14,15,16,17,18]. The reason for this might be that, although in the context of a developing country, the study was conducted within a well-organized facility with good sanitation standards compared to many surrounding areas. Proximity to a hospital and the possibility of medical treatment (e.g., antibiotics) in case of bacterial infections may have also played a role in identifying such a low number of affected subjects. However, our results have importance. In fact, besides the obvious impact on the “definite RHD” child, having identified 13 subjects with “borderline RHD” makes it possible to extend the six-monthly echocardiographic follow-up for them to act promptly with prophylactic therapy in case of the progression of valve disease (Figure 3).

From an echocardiographic perspective, our results confirm what is already known in the literature: RHD is predominantly an MV disease [5,6,11]. In our population, we were unable to identify any subjects with significant pathology of the AV or other heart valves. Given the choice of a young and asymptomatic population on which to conduct an RHD screening campaign, we did not highlight in our case series the characteristic lesions of rheumatic mitral valvulopathy typically described in the advanced stages, such as anterior leaflet doming with a classic “hockey stick” morphology, commissural fusion, and mitral stenosis. These findings are more commonly seen in the symptomatic phase of the disease, the prelude of the most feared cardiac and extracardiac complications (e.g., heart failure, atrial fibrillation, stroke), unfortunately present even at an early age.

Interestingly, we were able to diagnose six previously unknown congenital heart defects (3.7%): four atrial septal defects, one coronary artery fistula and one Patent Ductus Arteriosus. This high prevalence, if not due to chance, is difficult to explain. A possible explanation for this could be an expression of a few demographic and/or pathophysiological factors (i.e., sex between blood relatives, malnutrition, presence of extra-cardiac comorbidities as a possible expression of misrecognized syndromic pictures in the examined patients). In any case, the correct diagnosis of these disorders is an ancillary favorable effect of an RHD screening campaign and allows these individuals to be followed up in dedicated facilities, when available.

### Study Limitations

The monocentric nature of this study might affect the reproducibility of our results in other settings and/or population. Indeed, as already discussed, even if located in one of the poorest and most rural areas in the country, the enrollment site (a Ugandan orphanage with a larger access to health care facilities) may lead to an underestimation of RHD prevalence, also compared to the closest communities.

Even if able to provide the best RHD diagnostic accuracy, as well as other echocardiographic measurements, the WHO criteria applied in this study can be somehow subjective. However, in this study, two (or three) expert operators evaluated every examination, significantly lessening bias concerns.

## 5. Conclusions

In this study, we confirmed the feasibility of an RHD screening program in a selected Ugandan population of 163 children younger than 15 years old. Despite the low prevalence of “definite RHD” observed, we were still able to detect a significant number of “borderline RHD” and congenital heart disease cases early, with a positive impact on these patients’ health. This study supports the need of RHD screening campaigns in high-prevalence regions such as in sub-Saharan Africa.

## Figures and Tables

**Figure 1 children-09-01451-f001:**
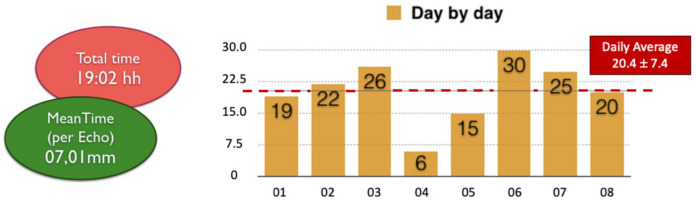
**General RHD screening data.** This figure reports the total time spent performing the echocardiographic screening on the whole population (red circle) and the mean time spent per exam (green circle). On the right side of the figure, a diagram shows the number of echocardiographic examinations performed per day. Abbreviations: RHD: rheumatic heart disease; hh: hours; mm: minutes.

**Figure 2 children-09-01451-f002:**
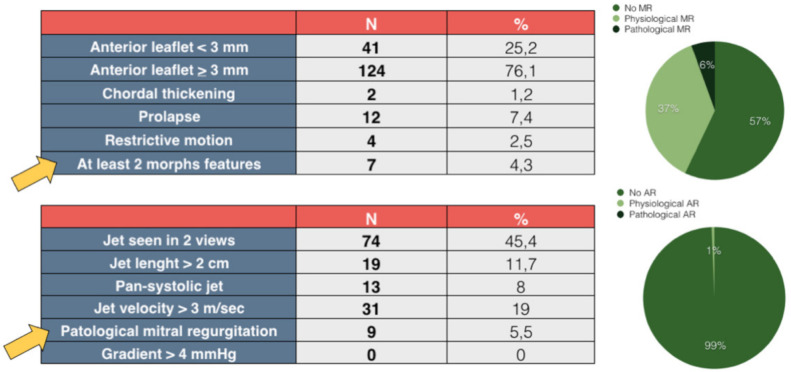
**Results of the RHD screening.** This figure summarizes the results of the RHD screening. Specifically, the top-left table shows the MV morphological features and the percentage of subjects showing at least 2 features (yellow arrow). The lower-left table shows the Doppler characteristics of MR jets and the percentage of patients who met the **“**pathological**”** MR definition (yellow arrow). In the upper-right diagram, the population is divided according to MR characteristics (none, physiological, pathological). The same division is made in the lower right for AR. Abbreviations: RHD: rheumatic heart disease; MR: mitral regurgitation; AR: aortic regurgitation; MV: mitral valve.

**Figure 3 children-09-01451-f003:**
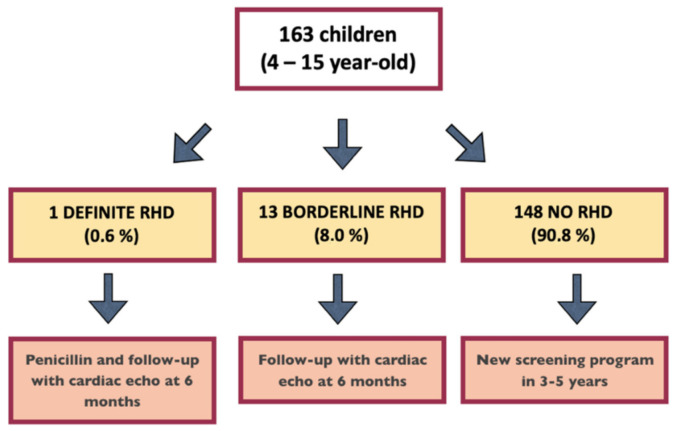
**RHD screening summary and follow-up advice.** The figure shows the results of the RHD screening, and the adopted follow-up strategy, as suggested by the WHO guidelines [8]. One patient is missing in the count since echocardiography was not feasible (see the Section 3. Results). Abbreviations: RHD: rheumatic heart disease.

**Figure 4 children-09-01451-f004:**
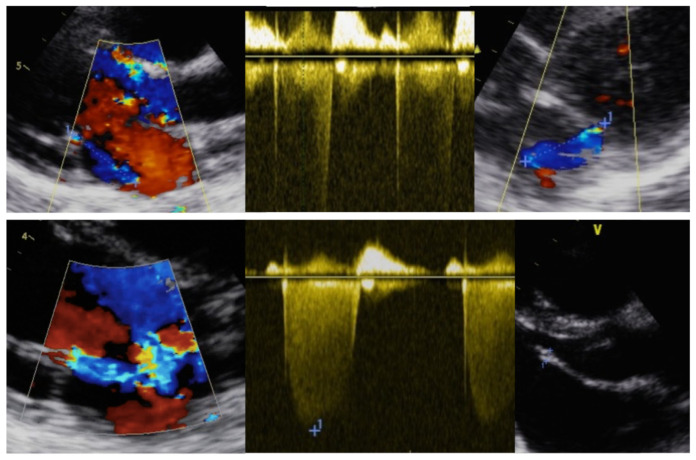
**Echocardiographic images of RHD subjects.** The top echocardiographic images show an example of a “borderline RHD” subject in which pathological MR is observed but without at least 2 morphological features of RHD. The bottom echo images show an example of a “definite RHD” individual; in this case, pathological MR plus 2 morphological features of RHD can be observed (MV anterior leaflet length > 3 mm and leaflet restrictive motion). Abbreviations: RHD: rheumatic heart disease; MR: mitral regurgitation; MV: mitral valve.

## Data Availability

Raw data will be available upon request on the website www.zenodo.org.

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
