# Peer review of "Echocardiographic Screening for Rheumatic Heart Disease in a Ugandan Orphanage: Feasibility and Outcomes"

_children, 2022, doi:10.3390/children9101451_

Round 1
Reviewer 1 Report
Rheumatic heart disease (RHD) is a major cause of acquired heart disease in developing countries. In this manuscript, the authors describe the results of an echocardiographic screening program for RHD carried out in a selected population of 163 non-symptomatic children living in an Ugandan orphanage. The aim of the study was to assess the feasibility of an echocardiographic RHD screening program in a locally confined population of a developing country.
Although the results of this study might be of interest to physicians working in developing countries, where RHD is a major cause of acquired heart disease, I miss some basic information about the study in the section "Materials and Methods". This is:
(1) The date or time period when the study has been conducted.
In this regards I wonder why you didn't include, in this manuscript, the data from the first follow-up examinations of the 13 borderline cases at 6 months after the first screening study? I think that inclusion of this data is of utmost importance for the evaluation of the feasibility of an effective screening program with the ultimate aim of preventing the progress of the disease in non-symptomatic patients.
(2) More detailed information about the geographic location (North-Uganda?) and the infrastructure of the orphanage. Although some information is given later in the Discussion ("facility with good sanitary standards compared to many surrounding areas; proximity to a hospital and the possibility of medical treatment"), I think that it is mandatory to present such information in M&M.
(3) Information about the organization which had planned and conducted the study and on the logistic background of the study. Such information is needed for evaluating the feasibility of a screening program in other populations. Discussion of logistic background is also missed in the Discussion section.
Please correct some typing errors (e.g. page 2, line 63/64 "children leaving in a North-Uganda orphanage") and wording (page 5; "subjects" may be replaced by a more convenient word, e.g. people or children).
Author Response
Darr Reviewer#1. We would like to really thank you for the time spent in reviewing our manuscript and your positive feedback.
We have revised the manuscript according to your comments and the one of the other reviewer. Our impression is that now the manuscript is significantly improved because of your suggestions.
Please see in the attached file a detailed point-by-point answer.
We remain at your complete disposal,
Best regards

Reviewer 2 Report
A very interesting, educational and well written manuscript that has clinical interest and merit. However, there are some editing issues that the authors should consider and address. The following are suggestions/comments regarding these issues. Line 27, "...hemodynamic complications have already occurred." Line 29, "...when appropriate the aim of the study ...". Line 31, "...the children (5-14 years old) living in a North-Uganda ...". Line 40, "...echocardiographic screening for RHD in an orphanage in Ugandan." Line 52, "...complications already have occurred [5]." Line 54, "...progressive fibrosis and consequential valvular damage [6,7]." Line 60, "The aim of the study is ...". Line 63, "...in all the children (5-14 years old) living in a North-Uganda orphanage." Line 68, "...diagnosis of RHD [8]." Lines 105-106, "...of the children presenting with pathological AR, while pathological MR was found ...". Line 120, "showing at least 2 features (yellow arrow)." Line 139, "...screening, and the adopted follow-up strategy as suggested by the WHO ...". Line 140, "...count since cardiac echocardiography was not ...". Line 154, "The top echocardiographic images show an ...". Line 155, "...subject in which pathological MR is observed ...". Line 156-157, "..."definite RHD" individual; in this case, pathological MR plus 2 ...". Line 166, "...significant data concerns is the short duration ...". Line 170, "...in a short time, thanks to increasingly ...". Line 173, "...echocardiographic examinations; for example, in an ...". Line 174, "...subjects are more frequently suffering, such as from clinically ...". Line 177, "...the need to create a report), in addition to ...". Lines 186-187, "...stroke (7.1%), or infective endocarditis (4%)." Lines 195 -196, "...younger patients diagnosed in the early stages of their disease have the most to gain from this earlier identification and institution of ...". Line 204, "...our results have importance. In fact, besides ...". Line 213, "...prevalence, if not due to chance, is difficult ...". Line 215, "...malnutrition, presence in the examined patients with extra-cardiac ...". Lines 225-226, "...underestimation of the RHD prevalence, also compared to ...". Lines 229-230, "...every examination, significantly lessening bias concerns."
Author Response
Darr Reviewer#2. We would like to really thank you for the time spent in reviewing our manuscript and your positive feedback.
We have revised the manuscript according to your comments and the one of the other reviewer. Our impression is that now the manuscript is significantly improved because of your suggestions.
Please see in the attached file a detailed point-by-point answer.
We remain at your complete disposal,
Best regards

Round 2
Reviewer 1 Report
Authors have successfully addressed all of my previous comments.